# Multi-Omics Analysis Reveals Synergistic Enhancement of Nitrogen Assimilation Efficiency via Coordinated Regulation of Nitrogen and Carbon Metabolism by Co-Application of Brassinolide and Pyraclostrobin in *Arabidopsis thaliana*

**DOI:** 10.3390/ijms242216435

**Published:** 2023-11-17

**Authors:** Ya-Qi An, De-Jun Ma, Zhen Xi

**Affiliations:** 1State Key Laboratory of Elemento-Organic Chemistry, Department of Chemical Biology, National Pesticide Engineering Research Center, College of Chemistry, Nankai University, Tianjin 300071, China; 1120180343@mail.nankai.edu.cn (Y.-Q.A.); madejun@nankai.edu.cn (D.-J.M.); 2Frontiers Science Center for New Organic Matter, Nankai University, Tianjin 300071, China; 3Collaborative Innovation Center of Chemical Science and Engineering, Tianjin 300071, China

**Keywords:** nitrogen assimilation efficiency, brassinolide, pyraclostrobin, transcriptome, metabolome

## Abstract

Improving nitrogen (N) assimilation efficiency without yield penalties is important to sustainable food security. The chemical regulation approach of N assimilation efficiency is still less explored. We previously found that the co-application of brassinolide (BL) and pyraclostrobin (Pyr) synergistically boosted biomass and yield via regulating photosynthesis in *Arabidopsis thaliana*. However, the synergistic effect of BL and Pyr on N metabolism remains unclear. In this work, we examined the N and protein contents, key N assimilatory enzyme activities, and transcriptomic and metabolomic changes in the four treatments (untreated, BL, Pyr, and BL + Pyr). Our results showed that BL + Pyr treatment synergistically improved N and protein contents by 56.2% and 58.0%, exceeding the effects of individual BL (no increase) or Pyr treatment (36.4% and 36.1%). Besides synergistically increasing the activity of NR (354%), NiR (42%), GS (62%), and GOGAT (62%), the BL + Pyr treatment uniquely coordinated N metabolism, carbon utilization, and photosynthesis at the transcriptional and metabolic levels, outperforming the effects of individual BL or Pyr treatments. These results revealed that BL + Pyr treatments could synergistically improve N assimilation efficiency through improving N assimilatory enzyme activities and coordinated regulation of N and carbon metabolism. The identified genes and metabolites also informed potential targets and agrochemical combinations to enhance N assimilation efficiency.

## 1. Introduction

Nitrogen (N) is necessary for plant growth and yields, as it is an essential constituent of key biological molecules such as amino acids, nucleic acids, chlorophyll, ATP, and phytohormones [1]. Plants primarily absorb N from the soil in the forms of nitrate (NO_3_^−^) and ammonium (NH_4_^+^) [2]. N metabolism in plants encompasses the processes of N acquisition, transport, and assimilation [2]. The absorption and transport of NO_3_^−^ relies on the low-affinity (NRT1) and high-affinity (NRT2) nitrate transporters and NH_4_^+^ acquisition is mediated by ammonium transporters (AMTs) as well as aquaporins or cation channels [3]. After uptake by the plant, inorganic N must be assimilated into amino acids to be utilized by plants [4]. This N assimilation process mainly involves the reduction in NO_3_^−^ to NH_4_^+^ and the incorporation of NH_4_^+^ into amino acids [5]. In the former process, NO_3_^−^ is reduced to nitrite (NO_2_^−^) by nitrate reductase (NR) in the cytosol, then NO_2_^−^ is translocated to the plastids and chloroplasts where it is further reduced to NH_4_^+^ by nitrite reductase (NiR) [5]. After that, NH_4_^+^ is directly assimilated into glutamate and glutamine through the glutamine synthetase (GS)/glutamate synthase (GOGAT) cycle, utilizing 2-oxoglutarate from the TCA cycle as carbon skeletons [5]. Subsequently, amino acids produced from N assimilation are transported as aspartic acid, glutamic acid, asparagine, and glutamine to serve as N donors for the subsequent biosynthesis of other N-containing compounds and secondary metabolites [6].

N assimilation is essential to plant growth and yields. Finding a method to improve nitrogen use efficiency (NUE) without affecting yield is an ongoing goal to maintain agricultural sustainability and food security [7]. In this way, it will be beneficial to increase plant productivity without adding production costs and environmental burdens from overusing N fertilizers [8]. Currently, genetic engineering approaches to overexpress genes involved in the N assimilation process have been developed to improve NUE and plant yields [9,10,11]. Overexpression of nitrate reductase (OsNR2) or nitrite reductase (OsNiR1) improved NUE and grain production in rice [12,13]. Similarly, overexpression of OsNADH-GOGAT1 in rice or TaNADH-GOGAT1−3B in wheat also enhanced NUE and grain yields [14,15]. Transgenic lines overexpressing glutamine synthetase (TaGS2−2Ab in wheat, ZmGLN1.3 in maize) also exhibited increased NUE and grain yields [16,17]. The constitutive overexpression of glutamine synthetase (OsGS1;2) in rice improved N assimilation and increased yields under particular conditions [18]. All the above evidence indicated the regulatory role of N assimilation in plant NUE and productivity. Since genetic breeding requires much time and sophisticated knowledge of gene function, trials to use small compounds to chemically regulate the N assimilation process for biomass and yield increases are also attractive but still less explored. 

As reported, small compounds including pyraclostrobin (Pyr, a widely used agricultural strobilurin fungicide) and brassinolide (BL, the most bioactive BR phytohormone) have been found to be capable of modulating N metabolism [19,20]. Exogenous application of Pyr has been observed to promote plant growth and development by enhancing NR activity via reducing cytoplasmic pH due to inhibition of the respiratory electron transport chain across multiple crops [21], such as soybean [22], maize [23], banana [24], tomato [25], and grafted cucumber plants [26]. Furthermore, Pyr treatment also enhanced the N harvest index by promoting the net remobilization of N from the vegetation organs to the reproduction organ in wheat [27] and tomato [28]. Foliar application of Pyr also enhanced nitrate assimilation of soybean by promoting rhizobial-nodule formation and nitrogenase activity [29]. Different from Pyr, BL was mainly involved in the responses to N deficiency [30]. On one hand, N deficiency promoted growth and root foraging responses by upregulating the transcription of components involved in BL signaling and biosynthesis, such as BAK1 (BL co-receptor protein) [31], BES1 and BZR1 (BL-activated transcription factor) [32,33], and BL biosynthetic genes (e.g., *DWF1*, *CPD*, *DWF4,* and *BR6OX2*) [34]. On the other hand, BL enhanced N acquisition and assimilation by modulating N regulators and transporters. BES1 induced the expression of high-affinity nitrate transporters-NRT2.1/2.2 by directly binding to their promoters, increasing nitrate uptake and N concentration [35]. BL treatment also promoted NH_4_^+^ uptake by stimulating ammonium transporters-AMT1;1 and AMT1;2 expression in rice [36]. Moreover, BL was also implicated in the N signaling pathway by activating HBI1 (a key regulator of the plant N signaling pathway) at the post-transcriptional level [37,38].

Beyond the positive effect of individual Pyr or BL on N assimilation that has been validated in diverse plants, we previously found a synergistic effect of the combined usage of BL and Pyr on boosting biomass and yield by promoting photosynthesis in *A. thaliana* [39]. It has been reported that the enhanced crop growth and yields rely on the coordinated N assimilation with the production of carbohydrates via photosynthesis [40,41]. This coordination is crucial as the N assimilation process supplies N-containing compounds essential for photosynthesis, such as photosynthesis enzymes and thylakoid N (proteins associated with bioenergetics and light-harvesting). Meanwhile, N assimilation itself requires photosynthesis to provide energy and organic carbon skeletons [42].Therefore, although the increase in biomass and yield by the combined usage of BL and Pyr was tightly associated with the enhanced photosynthetic process, it was still necessary to take the N assimilation process into consideration. However, the influence of the combined usage of BL and Pyr on N assimilation efficacy remained unclear.

For this purpose, we examined the N content and protein content, the activities of key N assimilatory enzymes (NR, NiR, GS, and GOGAT), transcriptional changes, and metabolic changes following the four treatments (untreated control, BL, Pyr, and BL + Pyr) in *A. thaliana*. We used the leaf as the research model, because leaves are the largest N sink of plant organs as well as the main site for N assimilation [43] and leaf N content shows positive correlation with photosynthetic efficiency [44,45,46,47,48]. Our results showed that the combined application of BL and Pyr exerted a synergistic effect on promoting N assimilation efficacy. This synergistic effect was attributed not only to the improved activities of key enzymes involved in the N assimilation process, but also to the coordinated regulation of processes related to N assimilation and carbon metabolism at the transcriptional and metabolic level. Our study indicated the feasibility of spraying a group of chemical compounds targeting diverse processes to promote N assimilation efficacy by coordinately regulating N metabolism and carbon metabolism. Additionally, the identified genes and metabolites potentially associated with enhanced N assimilation efficacy might facilitate the development of new genotypes and novel agrochemical combinations for enhanced NUE and yields.

## 2. Results

### 2.1. The Co-Treatment of BL and Pyr Improved N Content and Protein Content

To investigate whether the co-treatment of BL and Pyr also showed a synergistic effect on promoting N assimilation efficacy, we determined N and protein contents in *A. thaliana* leaves treated with BL, Pyr, and BL + Pyr, respectively. The measurements of N and protein concentrations revealed that the concentrations of leaf N and protein were more significantly increased by the BL + Pyr treatment than by the individual applications of BL and Pyr at all the above time points (Figure 1A,B). As shown in Figure 1A,B, the maximum increase in N and protein concentrations of the BL + Pyr group were 14.8% and 16.1%, surpassing the increases of 10.9% and 10.7% in the Pyr group, compared with the untreated group. In contrast, the leaf N and protein concentrations in the BL group were comparable to those in the untreated group. Then, based on their concentrations and the total dry weight of leaves per plant, we calculated the total weight of N and protein fixed in leaves per plant (Appendix A). As shown in Figure 1C,D, the BL + Pyr treatment fixed more N and protein in leaves per plant than the individual applications of BL and Pyr at all the above time points. The maximum increase in total weight of N and protein fixed in leaves per plant of the BL + Pyr group was 56.2% and 58.0%, exceeding the increases of 36.4% and 36.1% in the Pyr group, compared with the untreated group (Figure 1C,D). In contrast, BL did not improve the weight of total N and protein fixed in leaves per plant. Instead, a decrease of 9.5% and 10.3% due to the decreased leaf dry weight were observed in 39-day-old seedlings in the BL group, compared to the untreated group (Figure 1C,D). Notably, unlike the BL + Pyr group where the consistent improvements in both concentrations and total fixated content of leaf N and protein were found in seedlings from 27 to 39 days old, the Pyr group only exhibited significant increases in these parameters when the seedlings transitioned into the reproductive growth stage (31–39 days old), with no substantial effect during the vegetative stages (27 days old). Taken together, these results indicated that the co-application of BL and Pyr synergistically improved leaf N and protein contents.

### 2.2. The Enzyme Activity of NR, NiR, GS, and GOGAT Were All Enhanced by the Co-Application of BL and Pyr

To elucidate the underlying mechanism of the synergistic enhancement of leaf N and protein content by the co-application of BL and Pyr, we assessed N assimilation efficiency of the aforementioned four groups. This was achieved by measuring the activities of key enzymes involved in N assimilation, namely nitrate reductase (NR), nitrite reductase (NiR), glutamine synthetase (GS), and glutamate synthase (GOGAT).

As shown in Figure 2, the BL + Pyr treatment significantly increased the activities of all four enzymes, with the maximum increases in the activities of NR, NiR, GS, and GOGAT up to 354%, 42%, 62%, and 62% versus the untreated group, respectively. In contrast, the individual applications of BL and Pyr resulted in smaller improvements in the activities of these four N assimilation enzymes than the combined Pyr and BL treatment (Figure 2). Compared to the untreated group, the activities of NR, NiR, GS, and GOGAT in the Pyr group escalated to their peak by 104%, 31%, 32%, and 38%, respectively (Figure 2). In parallel, these enzyme activities of NR, NiR, GS, and GOGAT in the BL group showed a maximum elevation of 74%, 26%, 25%, and 28%, respectively (Figure 2). It is noteworthy that, in contrast to the consistent increases in N assimilation enzyme activities in seedlings from 27 to 39 days old by the BL + Pyr treatment, the significantly improved N assimilation enzyme activities in the Pyr group were only observed in the seedlings at the reproductive stage (31–39 days old) rather than at the vegetative stages (27 days old) (Figure 2). Furthermore, the significant augmentations in the activities of the N assimilation enzyme in the BL group were exclusively observed in seedlings at the vegetative development stage (27-day-old seedlings), with no substantial effect during the reproductive stage (31–39 days old), except for a decrease in NR and NiR by 39% and 27% in 39-day-old plants (Figure 2). These results suggested that the co-application of BL and Pyr resulted in a synergistic enhancement of N assimilation enzyme activities both in terms of magnitude and duration.

### 2.3. Genes Related to Protein Synthesis, Photosynthesis, and Carbohydrate Metabolism Were Uniquely Regulated by the Co-Application of BL and Pyr

We subsequently performed comparative transcriptomic analysis to reveal the molecular mechanisms underlying the synergistic enhancement of N assimilation efficacy by the co-application of BL and Pyr. RNA-Seq was performed on the leaves of 31-day-old seedlings, a time point at which a synergistic increase in the net photosynthetic rate was observed with the co-application of BL and Pyr [39]. Hierarchical clustering analysis of gene expression levels showed BL + Pyr elicited a unique transcriptional response distinct from the other three treatments (untreated, BL, and Pyr) (Figure 3A). By identifying the number of differentially expressed genes (DEGs), BL + Pyr differentially regulated 5823 genes (2917 up, 2906 down) versus the untreated control, 6618 genes (3291 up, 3321 down) versus BL treatment, and 3298 genes (1820 up, 1478 down) versus Pyr treatment, respectively (Figure 3B). Meanwhile, Pyr treatment altered 5717 genes (2680 up, 3031 down) and BL treatment altered 2718 genes (1454 up, 1264 down), compared to the untreated control (Figure 3B). To validate the reliability of the RNA-Seq data, the expression levels of 10 N metabolism-related DEGs were determined via qRT-PCR. As shown in Appendix A, the differential expression of these genes across four groups according to qRT-PCR was highly correlated with the RNA-seq data, confirming the reproducibility of the RNA-Seq data.

To explore whether the combination of BL and Pyr synergistically enhanced N assimilation efficacy by directly modulating the transcriptional patterns within the N metabolism pathway, we analyzed the DEGs annotated within this pathway. We found that the DEGs of N metabolism pathway in the BL + Pyr group (7 up, 22 down) and Pyr group (9 up, 24 down) were over-represented by downregulated genes, while these in the BL group (8 up, 1 down) were over-represented by upregulated genes (Figure 4). Furthermore, the co-application of BL and Pyr exhibited unique regulation of N transporters and assimilation enzymes at the transcriptional level compared to individual applications of BL or Pyr, following the subcellular localization and functions of proteins (Figure 4). As shown in Figure 4, BL + Pyr treatment exhibited differential regulation of high- and low-affinity nitrate transporters, upregulating high-affinity NRTs and downregulating low-affinity NRTs. In contrast, both up- and downregulated genes encoding high- or low-affinity nitrate transporters were found in the Pyr group. All the DEGs encoding nitrate transporters in the BL group were upregulated low-affinity NRTs. Notably, BL + Pyr treatment also regulated N-assimilating enzymes located in the chloroplasts and cytoplasm in different ways: through upregulating the genes that were encoding NiR and GS2 that were located in the chloroplast and downregulating the genes that were encoding NR, GS1, and NADPH-GOGAT that were located in the cytoplasm (Figure 4). In contrast, the DEGs encoding N-assimilating enzymes in the BL group were all upregulated, and those in the Pyr group were all downregulated (Figure 4). Additionally, the DEGs that were encoding glutamate dehydrogenase, acetamidases, cyanases, and nitrilases that catalyzed the conversion of glutamate, cyanate, formamide, and nitrile to ammonia were all downregulated by the combined BL and Pyr treatment, while individual applications of BL or Pyr resulted in both up- and downregulation of these genes (Figure 4). Collectively, these findings indicated an intricate and sophisticated transcriptional modulation of N metabolism genes orchestrated by the BL + Pyr treatment, in a manner depending on subcellular localization and function of genes.

However, the expression levels of genes in the N metabolism pathway did not fully correlate with the variations in N assimilation efficacy across the four groups. This suggested the BL + Pyr treatment might indirectly enhance N assimilation efficacy by transcriptionally regulating other interconnected processes. To elucidate the hub regulatory network underlying the synergistically enhanced N assimilation efficacy by the BL + Pyr treatment compared to individual applications of BL and Pyr, the DEGs uniquely regulated by the BL + Pyr treatment were further analyzed. These DEGs included the nonoverlapping DEGs between the BL + Pyr treatment and BL or Pyr treatments versus the untreated group and the overlapping DEGs between the BL + Pyr group and the other three groups (untreated, BL, and Pyr) (Figure 3C,D). As shown in Figure 3C,D there were 945 down- and 1048 upregulated genes in the BL + Pyr group versus the untreated control, respectively. Meanwhile, 455 up- and 546 downregulated genes were consistently differentially regulated by the BL + Pyr treatment versus the other three groups (untreated, BL, and Pyr). Analyzing these BL + Pyr-unique DEGs might inform the synergistic regulatory network potentiating N assimilation and N use efficiency by the BL + Pyr treatment.

GO and KEGG enrichment analyses were then performed on these BL + Pyr-specific DEGs, aiming at exploring the biological process and pathways uniquely regulated by the co-administered BL and Pyr. As shown in Figure 5A and Appendix A, the most significantly enriched GO terms of those DEGs were related to protein synthesis, e.g., organic N compound biosynthetic processes, peptide biosynthetic processes, amide biosynthetic processes, cellular amino acid biosynthetic processes, ribosome assembly, translation, ribonucleoprotein complex biogenesis, protein folding, etc., belonging to the classification of biological processes; the ribosome and polysome in the category of cellular components; poly(A) RNA binding, unfolded protein binding and translation in the category of molecular functions. In addition, these DEGs were also significantly enriched in photosynthesis-related GO entries (e.g., chloroplast, photosynthesis, carbon fixation, chlorophyll biosynthetic process, and photorespiration) (Figure 5A and Appendix A). Notably, the significantly enriched GO terms involved in protein synthesis and photosynthesis were both over-represented by upregulated DEGs. Meanwhile, the significantly enriched GO terms were also implicated in the metabolism of organic acids, including carboxylic acid biosynthetic and metabolic processes, pyruvate metabolic processes, gluconeogenesis, glycolytic processes, carbohydrate catabolic processes, ATP generation from ADP, and the generation of precursor metabolites, energy, and mitochondrion (Figure 5A and Appendix A).

Similarly, the KEGG enrichment analysis also showed these BL + Pyr-specific DEGs were enriched in the pathways for protein synthesis (the biosynthesis of amino acids, valine, leucine, and isoleucine biosynthesis, and amino acid metabolism), photosynthesis (porphyrin metabolism and carbon fixation in photosynthetic organisms), and carbohydrate metabolism (C5-Branched dibasic acid metabolism, 2-oxocarboxylic acid metabolism, glycolysis/gluconeogenesis, carbon metabolism, and pyruvate metabolism) (Figure 5B and Appendix A). Notably, the most significant enrichment in the KEGG pathway was ribosome, with all of 121 DEGs annotated to this pathway upregulated (Figure 5B). These results suggested that the BL + Pyr treatment uniquely increased the transcription of genes related to protein synthesis and photosynthesis, while also modulating the transcriptional patterns of carbohydrate metabolism.

To identify the hub genes of the transcriptional network regulated by the combined application of BL and Pyr, we specially screened the transcription factors that were differentially regulated by BL + Pyr compared to individual BL or Pyr treatments for in-depth analysis. There were 133 transcription factor families belonging to 34 transcription factor families that were only differentially regulated by co-administration of BL and Pyr rather than by the administrations of BL and Pyr alone (Appendix A and Table 1). Literature mining revealed that 36 of those 133 BL + Pyr-specific transcription factors played a role in N metabolism [49], belong to the families of ERF, WRKY, RAV, bZIP, ARF, C2H2 zinc-finger, HD-ZIP, MYB, NAC, GATA, GRAS, GeBP, HSF, MIKC_MADS, SBP, TCP, and ZF-HD (Appendix A and Table 1). These 36 BL + Pyr-unique transcription factors might be the key nodes controlling the synergistically increased NUE in response to the BL + Pyr treatment. However, the remaining 97 BL + Pyr-specific transcription factors not previously linked to N metabolism belonged to the families with members that have been reported to govern N metabolism, including the families of bHLH (11), MYB_related (7), C2H2 zinc-finger (6), MYB (6), B3 (4), C3H zinc-finger (4), ERF (4), G2-like (4), NAC (4), Trihelix (4), WRKY (4), bZIP (4), CO-like (3), Dof (3), TCP (3), DBB (2), GRAS (2), MIKC_MADS (2), SBP (2), ARF (1), GeBP (1), HSF (1), TALE (1), and ZF-HD (1) (Appendix A and Table 1). Furthermore, the other 13 transcription factors belonged to the families of ARR-B (4), BTB/POZ (4), BBR/BPC (1), FAR13 (1), SRS (1), YABBY (1), and PLATZ (1) (Appendix A and Table 1). These 97 BL + Pyr-specific transcription factor might enhance NUE in a novel manner, warranting further investigation.

### 2.4. The Metabolites Related to N Metabolism and Carbon Metabolism Were Coordinately Regulated by the Co-Application of BL and Pyr

Metabolomic analysis was subsequently performed to identify metabolites associated with the synergistically improved N assimilation efficacy by the combined application of BL and Pyr. Based on the principal component analysis (PCA) analysis, the accumulation of secondary metabolites in the BL + Pyr group was distinct from that in all of the other three groups (untreated, and individual BL and Pyr treatments) (Figure 6A). Differentially accumulated metabolites (DAMs) analysis revealed that the BL + Pyr treatment differentially regulated 232 metabolites (94 increased, 138 decreased) versus the untreated control, 228 metabolites (101 increased, 127 decreased) versus BL individual treatment, and 71 metabolites (18 increased, 53 decreased) versus Pyr individual treatment, respectively (Figure 6B). Meanwhile, individual Pyr treatment altered 204 metabolites (82 increased, 122 decreased) and individual BL treatment altered 229 metabolites (77 increased, 152 decreased), compared to the untreated control (Figure 6B). These results indicated that the BL + Pyr treatment elicited substantial metabolic reprogramming compared to individual BL and Pyr treatments.

We then analyzed the accumulations of the intermediates annotated to the N assimilation pathway to explore whether the combination of BL and Pyr synergistically enhanced N assimilation efficacy by directly modulating the metabolites implicated in N assimilation. The BL + Pyr treatment elicited distinct effects on the metabolic profiles of the N assimilation pathway from BL or Pyr treatments (Figure 7). As shown in Figure 7, the co-application of BL and Pyr increased the contents of glutamine and α-ketoglutarate contents by 44% and 46%, exceeding increases by Pyr alone (27% and 29%) and no change by BL alone, compared with the untreated group. Conversely, the level of asparagine in the BL + Pyr group was reduced by 21% but did not significantly change by individual BL and Pyr applications, compared to the untreated group. Meanwhile, the level of alanine was only significantly decreased in the BL groups (17%), but did not change in the BL + Pyr and Pyr groups, compared to the untreated control. In addition, glutamate and aspartate contents were not significantly varied across the four groups. Furthermore, we compared the metabolic profiles of amino acids and their derivatives, to explore whether the combination of BL and Pyr also influenced amino acid metabolism at the metabolite level. As shown in Appendix A, the accumulations of amino acids and their derivatives in the BL + Pyr group were also different from those of the remaining three groups (untreated, BL, and Pyr). These results suggested that the BL + Pyr treatment uniquely modulated metabolites connected to the N assimilation process and amino acid metabolism.

To investigate whether the coordinated regulation of metabolites across interconnected pathways also contributed to the synergistic enhancement of N assimilation efficacy by combining BL and Pyr, we mapped the DAMs to the KEGG pathway for further analysis. As shown in Appendix A, there are four pathways uniquely enriched in the BL + Pyr group rather than in the BL and Pyr groups, including photosynthesis, inositol phosphate metabolism, vitamin B6 metabolism, and monoterpenoid biosynthesis. In addition, the remaining pathways enriched in the BL + Pyr group were highly overlapped with those enriched in the BL and Pyr individual groups. On the one hand, the enriched pathways related to photosynthesis in the BL + Pyr group were in line with those observed in the BL group rather than the Pyr group, such as carbon fixation in photosynthetic organisms and nicotinate and nicotinamide metabolism, etc. On the other hand, the enriched pathways associated with the metabolism of N-containing compounds and organic acids in the BL + Pyr group were observed in the Pyr group rather than the BL group, such as N metabolism, TCA cycle, starch and sucrose metabolism, C5-branched dibasic acid metabolism, glyoxylate and dicarboxylate metabolism, pentose and glucuronate interconversions, etc. These results suggested that the co-treatment with BL and Pyr coordinately regulated metabolites associated with photosynthesis and those linked to N and carbon metabolism, which was not achieved by the BL and Pyr individual applications.

Afterward, we conducted a detailed analysis of the metabolites that differentially accumulated in the BL + Pyr group rather than in the individual BL and Pyr groups. It could further identify the metabolites related to the synergistic improvement of N assimilation efficacy unique to the co-application of BL and Pyr. As shown in Figure 6C,D, 57 DAMs were fitting the above criteria based on Venn diagram analysis, with 38 decreased and 19 increased. As shown in Table 2, these 57 DAMs span 14 compound classes. The top classes included 10 flavonoids and derivatives, 7 carbohydrates and derivatives, 6 amino acids and derivatives, and 5 terpenoids. Additional classes with three or fewer DAMs were indoles and derivatives, nucleotides and derivatives, alkaloids and derivatives, amines, glycerolipids, organic acid and its derivatives, fatty acyls, cinnamic acids, and derivatives, phenylpropanoids, and benzoic acids and derivatives (Table 2). Notably, 11 of these 57 BL + Pyr-specific DAMs achieved >2-fold change versus the untreated group (Table 2). Among the 11 DAMs, except for Denin and aspartic acid di-O-glucoside that increased four-fold and six-fold, respectively, the remaining DAMs all showed a significantly decreased accumulation (Table 2). For instance, 3-indoleacetonitrile and (-)-menthol decreased nearly 13-fold. Other metabolites, listed in descending order of fold-change reduction versus the untreated group, included L-gulono−1,4-lactone, scopolamine, methyl vanillate, orotic acid, N-p-coumaroylputrescine, Dihydroxy tomatidine-O-hexosyl-O-rhamnoside, and indole−3-carboxaldehyde (Table 2). Furthermore, there were seven of these 57 BL + Pyr-specific DAMs consistently differentially regulated by BL + Pyr versus the other three groups (untreated, BL, and Pyr groups), including two that increased in abundance (Indole−3-acetamide and L-arginine) and five that decreased in abundance (monopalmitin, dihydroartemisinic acid, indole−3-carboxaldehyde, d-glucose, and (-)-menthol) (Table 3).

The 57 BL+ Pyr-specific DAMs were further mapped to the KEGG pathway to explore the metabolic pathways uniquely regulated by the co-administered BL and Pyr compared to the BL and Pyr groups. As shown in Figure 8, 20 of these 57 DAMs were annotated, distributing across 25 pathways. The majority of those 25 pathways were implicated in carbohydrate metabolism, photosynthesis, and amino acid metabolism. Specifically, the most significantly enriched pathway was monoterpenoid biosynthesis. The remaining annotated pathways also included the pathways related to carbohydrate metabolism, e.g., pentose phosphate pathway, starch and sucrose metabolism, fructose and mannose metabolism, amino sugar and nucleotide sugar metabolism, glycolysis/gluconeogenesis, etc. They also involve pathways concerning photosynthesis (e.g., photosynthesis and carbon fixation in photosynthetic organisms), and the pathways of amino acid metabolism (e.g., tryptophan metabolism, arginine biosynthesis, and phenylalanine, tyrosine and tryptophan biosynthesis, etc.). In addition, we found that Adenosine 5′-Diphosphate, D-glucose, D-erythrose 4-phosphate, and D-glucose 6-phosphate were simultaneously annotated to multiple KEGG pathways, suggesting potential importance as integrative regulatory hubs altering metabolite accumulation in the BL + Pyr-specific network (Figure 8). These results showed that the BL + Pyr co-treatment uniquely reprogrammed the metabolite accumulation in the pathways related to N metabolism and carbon metabolism.

### 2.5. The Accumulation of DAMs Was Highly Correlated with the Regulation of DEGs

Pearson correlation analysis was subsequently performed between all DEGs and DAMs across these four groups to evaluate the correlation between the difference in metabolic and transcriptional profiles of the four groups (untreated, BL, Pyr, and BL + Pyr). As shown in Appendix A, the accumulation of DAMs was highly aligned with the expression levels of DEGs across these four groups. Among 483 total DAMs and 10,592 total DEGs of these four groups, 99 DAMs and 6309 DEGs were highly correlated (r > 0.99, *p* < 0.01), comprising 8081 negative and 8607 positive correlation pairs (Appendix A). This result indicated that the changes in the accumulation of metabolites among these four groups were closely associated with the regulation of transcriptional networks.

All DEGs and DAMs across these four groups were then mapped to KEGG pathways and the commonly enriched pathways were further screened to identify the metabolism pathways governing the crosstalk between transcriptional and metabolic networks. We found that the majority of highly enriched metabolism pathways shared by DEGs and DAMs were involved in amino acid metabolisms and protein synthesis, such as aminoacyl-tRNA biosynthesis, biosynthesis of amino acids, arginine biosynthesis, valine leucine and isoleucine biosynthesis, alanine aspartate and glutamate metabolism, cysteine and methionine metabolism (Appendix A). In addition, these DEGs and DAMs were also commonly enriched in porphyrin and chlorophyll metabolism as well as 2-oxocarboxylic acid metabolism, which were essential for photosynthesis and carbohydrate metabolism, respectively (Appendix A). The results suggested that the regulation of amino acid metabolism and protein synthesis, photosynthesis, and carbohydrate metabolism could serve as pivotal components in the interplay between the differential transcriptional responses and metabolic alterations elicited by these 4 treatments.

To pinpoint the hub DEGs and DAMs as well as the pathways governing the crosstalk between the transcription response and metabolic response uniquely elicited by the BL + Pyr treatment, we further performed the combined analysis focused on the genes and metabolites specifically differentially regulated by the BL + Pyr treatment rather than by the BL or Pyr alone. Based on the Pearson correlation analysis, out of 2465 DEGs and 57 DAMs specific to BL + Pyr, 1900 DEGs and all 57 DAMs were highly correlated (r > 0.99, *p* < 0.01), including 2378 negative and 2365 positive pairs (Appendix A). This result indicated a strong correlation between the regulation of transcription specific to the BL + Pyr treatment and the changes in metabolite content specific to the BL + Pyr treatment. Then, BL + Pyr-specific DEGs and DAMs were mapped to KEGG pathways for further analyzing the commonly enriched pathways. As shown in Figure 9, there were 20 KEGG pathways shared by BL + Pyr-specific DEGs and DAMs. The most abundant co-enriched pathways were those implicated in carbohydrate metabolism (pentose phosphate pathway, starch and sucrose metabolism, fructose and mannose metabolism, glycolysis/gluconeogenesis, pentose and glucuronate interconversions, amino sugar, nucleotide sugar metabolism, and galactose metabolism), followed by those related to amino acid metabolism (tryptophan metabolism, arginine biosynthesis, tyrosine metabolism, and phenylalanine metabolism) and those involved in the photosynthetic process (photosynthesis, oxidative phosphorylation, and carbon fixation in photosynthetic organisms). In addition, the co-enriched pathways also included inositol phosphate metabolism, biosynthesis of unsaturated fatty acids, vitamin B6 metabolism, ascorbate and aldarate metabolism, zeatin biosynthesis, and pyrimidine metabolism. These results highlighted the vital role of the unique regulation of carbohydrate metabolism, amino acid metabolism, and photosynthesis in the crosstalk between the transcriptional responses and metabolic alterations uniquely induced by the BL + Pyr treatment. The BL + Pyr-unique DEGs and DAMs annotated to the shared KEGG pathways (Appendix A) might be the hub regulators in the interaction between the transcription response and metabolic response specifically induced by the BL + Pyr treatment.

## 3. Discussion

Improving N assimilation efficiency is one of the key approaches to confer high NUE, which is essential for sustainable agriculture and food security [2,9,10]. However, achieving this goal requires coordinated regulation of plant growth, N assimilation, and carbon fixation [40,41], posing challenges for breeding high-yield genotypes with the improved NUE via genetic engineering technology [50]. In contrast, directly applying a group of chemical compounds targeting different processes may represent a promising approach to coordinately regulate plant growth, N metabolism, and carbon metabolism, and thus improve NUE accompanied by yield increases.

Inspired by the synergistic effects on boosting plant biomass and yield by improving photosynthesis efficacy with the co-application of BL and Pyr in our previous work [39], we further explored the effect on N assimilation efficiency with the co-application and individual applications of BL and Pyr. Our results demonstrated that the co-application of BL and Pyr also exerted a synergistic effect on improving the leaf N and protein concentrations as well as the total N and protein content fixed in leaves per plant compared to the individual BL and Pyr applications in *A. thaliana*. The co-increased photosynthesis efficacy and leaf N content by the co-application of BL and Pyr was consistent with the positive relationship between the photosynthetic performance and leaf N content in previous literature: the higher N content of leaves enabled the high-yield rice cultivar to maintain a high photosynthesis rate [45]; excess N accumulation in maize leaves was beneficial to photosynthesis and yield [46]; and leaf N content was positively correlated with leaf and panicle photosynthesis efficacy at the anthesis stage of rice [47]. Given uniform N supply across the four groups, it was reasonable to speculate that the combined application of BL and Pyr synergistically enhanced N assimilation efficacy compared to individual applications of BL and Pyr.

Nitrate reductase (NR), nitrite reductase (NiR), glutamine synthetase (GS), and glutamate synthase (GOGAT) play key roles in determining the N assimilation efficacy [5], of which the NR-catalyzed reduction in NO_3_^−^ is the rate-limiting step for efficiency of N acquisition and utilization [51]. Extensive evidence confirmed that enhancing the activities of key enzymes involved in N assimilation was capable of improving NUE and thus plant biomass and yield [2,9,10]. In this study, the co-application of BL and Pyr synergistically increased the activities of NR, NiR, GS, and GOGAT, surpassing the effects observed with BL or Pyr alone treatments both in terms of magnitude and duration. This result suggested that the synergistically enhanced NUE by the co-application of BL and Pyr was associated with improved N assimilation efficiency. This conclusion was further supported by the higher accumulations of glutamine and α-ketoglutarate with the co-application of BL and Pyr than the individual applications of BL and Pyr, as the contents of glutamine and α-ketoglutarate, the key intermediates in the N assimilation process, largely reflected N assimilation efficiency [5,6]. Given that the GS/GOGAT cycle was responsible for assimilation of NH_4_^+^ to glutamine and glutamate using α-ketoglutarate from TCA cycle as carbon skeleton, and thus linking N and carbon metabolism [52], the synergistically increased activities of GS and GOGAT as well as α-ketoglutarate levels with BL + Pyr co-treatment implied that the synergistical NUE enhancement might be also associated with the coordinated regulation of N and carbon metabolism beyond directly promoting N assimilation efficiency.

This hypothesis is further substantiated by transcriptome and metabolome analyses. In contrast to the general upregulation trend of the N metabolism pathway in the BL group, the BL + Pyr treatment transcriptionally modulated the N metabolism pathway in a manner dependent on subcellular localization and function of genes. For instance, the BL + Pyr treatment upregulated high-affinity NRTs and downregulated low-affinity NRTs as well as upregulated the NiR and GS2 located in the chloroplast and downregulated the NR, GS1, and NADPH-GOGAT that located in the cytoplasm [3]. This discrepancy in the expression levels of genes in the N metabolism pathway did not entirely align with the differences in N assimilation efficiency among different groups, suggesting synergistically enhanced N assimilation efficiency by the co-application of BL and Pyr did not rely solely on the direct transcriptional activation of the N metabolism pathway. Instead, it might indirectly enhance N assimilation efficiency by regulating other interconnected processes.

Photosynthesis and carbohydrate catabolism provide energy for N metabolism, amino acid, and protein synthesis, which are the primary modes of N utilization in plants [6,42]. Meanwhile, the carbon skeletons required for amino acid synthesis derive from carbon metabolism, such as glycolysis, carbon fixation in photosynthesis, oxidative pentose phosphate pathways, tricarboxylic acid cycles (TCAs), etc. Thus, the synthesis of amino acids closely links carbon metabolism to N metabolism [53]. Subsequent analysis of BL + Pyr-specific DEGs based on RNA-Seq revealed that, BL + Pyr synergistically activated genes related to amino acid, peptide, and protein synthesis, as the most pronounced transcriptional changes unique to BL + Pyr treatment were the upregulation of genes involved in protein synthesis and amino acid synthesis. Concurrently, BL + Pyr treatment also significantly upregulated photosynthesis-related genes and altered the transcriptional profiles of carbohydrate metabolism. These results provide further evidence that the combined BL + Pyr treatment coordinately regulates nitrogen metabolism, protein and amino acid biosynthesis, carbon utilization, and photosynthesis at the transcriptional level. This coordinated regulation not only enhanced nitrogen assimilation efficiency but also promoted N incorporation into peptides and proteins.

Furthermore, metabolomic analysis revealed coordinated regulation of metabolites associated with photosynthesis and those linked to N and carbon metabolism by the combined BL + Pyr treatment. This coordinated regulation was not observed with individual BL or Pyr applications. KEGG enrichment analysis of DAMs showed overlapping regulation of photosynthetic pathways by BL + Pyr and BL alone, and overlapping effects on nitrogen and carbohydrate metabolism by BL + Pyr and Pyr alone. Notably, the BL + Pyr combination uniquely modulated the photosynthesis pathway. Similarly, KEGG mapping of the BL + Pyr-specific DAMs further supported that the BL and Pyr co-treatment uniquely reprogramed the metabolite accumulation of key pathways governing carbon flux, photosynthetic output, and N assimilation compared to the individual BL or Pyr treatments.

Integrated transcriptomic and metabolomic analysis further highlighted the coordinated modulation of amino acid metabolism, protein synthesis, photosynthesis, and carbohydrate metabolism as central to the interplay between transcriptional and metabolic changes induced by the four treatments. Additionally, the regulation of carbohydrate metabolism, amino acid metabolism, and photosynthesis mediated the crosstalk between transcriptional and metabolic profiles uniquely altered by BL + Pyr compared to individual BL or Pyr treatments. Therefore, the synergistic improvement of nitrogen assimilation efficiency by BL + Pyr might be attributed to coordinated regulation of nitrogen metabolism, protein synthesis, photosynthesis, and carbohydrate metabolism. This coordinated regulation also enhancing N assimilation product utilization via promoting amino acid and protein biosynthesis.

Modulating regulators that control the coordination between carbon assimilation and N utilization represents a promising approach to enhance N assimilation efficiency and crop productivity, but the identification of such regulators is still underway [40,41]. Analysis of BL + Pyr-unique transcription factors might be helpful for identifying transcriptional regulators controlling the coordination between carbon assimilation and N utilization, as the co-application of BL and Pyr displayed a synergistic effect on boosting plant biomass and yield by the coordinated regulation of N metabolism and carbon metabolism [39]. In this work, 133 transcription factor families belonging to 34 transcription factor families have been identified to be uniquely regulated by the co-application of BL and Pyr rather than by the individual applications of BL and Pyr. Literature mining revealed that 36 of those 133 BL + Pyr-specific transcription factors were involved in N metabolism and response [49], while the remaining 97 BL + Pyr-specific transcription factors were not previously confirmed to be linked to N metabolism. Furthermore, we also identified 57 secondary metabolites spanning 14 compound chasses that were differentially regulated only by the co-application of BL and Pyr but not by the individual applications of BL and Pyr. Further in-depth analysis of these 97 BL + Pyr-unique transcription factors and 57 BL + Pyr-unique DAMs might be helpful for unveiling new genetic and chemical targets to enhance NUE and thus yield. Based on the joint analysis of the transcriptome and metabolome, we identified 20 KEGG pathways shared by BL + Pyr –specific DEGs and DAMs. The BL + Pyr-specific DEGs and DAMs enriched in these shared pathways might also be the potential candidate genes and compounds for improving N assimilation efficiency. Analyzing metabolites and fluxes within these shared pathways represents promising directions for future research.

In summary, we revealed a synergistic effect of the combination of BL and Pyr on improving N assimilation efficiency in *A. thaliana*. BL + Pyr treatment improved N and protein contents by 56.2% and 58.0%, which was associated with enhanced N assimilatory enzyme activities and the coordinated regulation of N and carbon metabolism. On one hand, the combination of BL and Pyr achieved a synergistic enhancement of N assimilation efficiency by increasing key N assimilatory enzyme activities of NR, NiR, GS, and GOGAT by 354%, 42%, 62%, and 62%, respectively. On the other hand, the co-application of BL and Pyr improved N assimilation efficiency through coordinated regulation of N assimilation, peptide and amino acid synthesis, photosynthesis, and carbohydrate metabolism at both transcriptional and metabolic levels. This synergistic enhancement of N assimilation efficiency resulted in boosted N content. Furthermore, the synergistically enhanced peptide and amino acid synthesis by the co-application of BL and Pyr also promoted the conversion of N assimilation products into peptide and protein in plant, thereby increasing the protein content. This work will provide detailed molecular evidence of applying a combination of compounds that target diverse processes to increase N assimilation efficiency via the coordinated regulation of N metabolism and carbon metabolism. The identified transcription factors and metabolites might reveal novel targets for genetic modifications or chemical manipulation of gene networks regulating N assimilation efficiency for biomass and yield increase.

## 4. Materials and Methods

### 4.1. Plant Material and Experimental Design

Seeds of *A. thaliana* (Col−0) were surface sterilized in 70% ethanol for 10 min, and washed with distilled water, then plated on 1/2 Murashige and Skoog medium supplemented with 1% sucrose and 0.8% agar. After stratification at 4 °C for 2 days, plates were placed horizontally in the artificial climate chamber under long-day conditions (16 h light/8 h dark, 22 °C, 30% humidity, and 120 photons μmol m^−2^ s^−1^) for 7 days. Then, seedlings were transferred to 7cm square pots filled with a peat-based commercial potting soil mixture (0–5 mm, KLASMANN, Germany) with added vermiculite (2–4 mm), at a volume ratio of 2:1 (peat soil to vermiculite). Each pot, containing one seedling, was sub-irrigated with tap water as needed, with no additional fertilization provided. After 14 days of growth in the artificial climate chamber under the same long-day conditions, uniform seedlings were selected and divided into four groups. Each group containing 20 biological replicates was sprayed with one of the following solutions: 1‰ DMSO, pyraclostrobin (Pyr, Shanghai Lvze Bio-Tech Co. (Shanghai, China) ≥99%) alone, brassinolide (BL, J&K SCIENTIFIC LTD (Beijing, China), ≥98%) alone, or a mixture of BL with Pyr. Each seedling received a spray volume of 2 mL, with treatments applied only once. The seedlings treated with 1‰ DMSO were considered as an untreated control group. BL and Pyr concentrations were set as 1 μM and 3 μM, respectively, at which a synergistic increase in photosynthetic efficiency and biomass with BL + Pyr co-treatment was previously demonstrated [39]. Leaves were harvested at 27, 31, and 39 days after sowing for physiological and biochemical analyses, as those time point spanned the stage when BL + Pyr treatment was observed to synergistically enhance photosynthetic efficiency in our previous study [39].

### 4.2. Measurement of Leaf N Content and Protein Content

Leaf N content and protein content of seedlings aged 27, 31, and 39 days in the four groups (untreated control, BL-treated, Pyr-treated, and BL + Pyr-treated) were determined using the Kjeldahl method, respectively [54]. Specifically, leaves were heated at 105 °C for 20 min and dried for 48 h at 75 °C, followed by grounding in a Wiley mill to pass through a 20-mesh screen. To determine protein N concentration, ~0.2 g powdered leaves were weighed and added with 5 mL of 5% trichloroacetic acid for protein extraction. After immersing in a 90 °C water bath for 15 min, the extraction solution was centrifuged at 4000 rpm for 15 min to isolate precipitates containing protein N. Then, the precipitates were washed with trichloroacetic acid, and dried at 50°C, before subjecting to protein N measurement. Meanwhile, dried, ground leaf samples without pre-treatment were prepared for total N concentration measurement. Afterward, the prepared leaf samples for total N and protein N concentration measurements were digested in concentrated H_2_SO_4_ with CuSO_4_ as a catalyst at 420 °C for 2 h. After cooling, the digest was diluted with distilled water, before transferring to the distillation apparatus along with an aliquot of NaOH to convert ammonium to NH_3_ gas. Then, the liberated NH_3_ was distilled into a trapping solution of boric acid and titrated with a standardized HCl solution to determine N concentration. Protein concentration was calculated as follows: Crude protein (% DW) = Protein N (%) × 6.25. Total N content and protein N content of leaves per plant were calculated as the dry weight of leaves per plant multiplied by total N concentration and protein concentration, respectively. Three technical replicates were performed for each treatment and 3 independent experiments were performed to obtain the results.

### 4.3. Measurement of N Assimilating Enzyme Activity

The activities of N-assimilating enzymes were performed on the leaves harvested from plants aged 27, 31, and 39 days in the four groups (untreated control, BL-treated, Pyr-treated, and BL + Pyr treated), respectively.

#### 4.3.1. Measurement of Nitrate Reductase (NR) Enzyme Activity

NR activity was measured following a published protocol [55]. Briefly, ~500 mg fresh leaf samples were homogenized in a chilled mortar with 4 mL of 25 mM phosphate buffer (pH 8.7) containing 10 mM cysteine and 1 mM EDTA at 4 °C. The homogenate was centrifuged at 10,000 g for 15 min at 4 °C and the supernatant was collected to measure NR activity. Then, we incorporated 0.40 mL of the supernatant into a 1.2 mL reaction solution. This solution was formulated with 1.2 mL of 0.1 M phosphate buffer (pH 7.5) supplemented with 0.1 M KNO_3_, and 0.4 mL of 3 mM NADH. The chemical interaction within this mixture was sustained for 30 min at 30 °C to produce nitrite ions (NO_2_^−^). Afterwards, the content of NO_2_^−^ was quantified by diazotization with 1 mL of 1% sulfanilamide in 3 M HCl and 1 mL of 0.02% N-naphthyl-ethylenediamine, followed by centrifugation at 10,000 rpm for 10 min. The absorbance of the supernatant was recorded at 540 nm. NR activity was calculated by comparing it with a standard curve of NaNO_2_. The unit of NR enzyme activity (U) expressed as μmol NO_2_^−^ produced per gram fresh weight per hour (U = nmol NO_2_^−^ g^−1^ h^−1^ FW). Each independent experiment consisted of three biological replicates and three independent experiments were performed to obtain the results.

#### 4.3.2. Measurement of Nitrite Reductase (NiR) Enzyme Activity

Nitrite reductase (NiR) activity was determined according to a published protocol with some modifications [56]. A total of ~500 mg fresh leaf samples were homogenized in an extraction buffer (pH 7.5) containing 100 mM HEPES, 5 mM cysteine, and 1 mM EDTA. Then, the homogenate was centrifuged at 4000 g for 5 min at 4 °C and the supernatant was used as the enzyme extract. The reaction mixture consisted of 100 mM HEPES (pH 7.5), 1 mM NaNO_2_, 0.2 mM methyl viologen, 10 mM sodium dithionite, and 200 μL enzyme extract in a total volume of 3 mL. The reaction was initiated by adding sodium dithionite and incubated at 25 °C for 30 min in the dark. After incubation, the reaction was stopped by adding p-Aminobenzenesulfonic acid. Then, the mixture was centrifuged at 10,000 rpm for 10 min and the absorbance of the supernatant was measured at 540 nm. NiR activity was calculated from a standard curve of NaNO_2_ and expressed as μmol NO_2_^−^ reduced per gram fresh weight per hour. Each independent experiment consisted of 3 biological replicates and 3 independent experiments were performed to obtain the results.

#### 4.3.3. Measurement of Glutamine Synthetase (GS) Enzyme Activity

GS activity was assayed according to a published protocol [55], with some modifications. Briefly, ~500 mg frozen leaf samples were homogenized in a 6 mL extraction buffer (pH 8.0) containing 50 mM Tris–HCl, 2 mM MgCl_2_, 2 mM DTT, and 0.4 M sucrose at 4 °C. The homogenate was centrifuged at 15,000 g for 20 min at 4 °C and the supernatant was used as enzyme extract. To initiate the reaction, 0.7 mL of the enzyme extract was introduced into the reaction mixture. This mixture was composed of 0.7 mL of 40 mM ATP, combined with 1.6 mL of 0.1 M Tris-HCl buffer (pH 7.4). The buffer contained 20 mM Na-glutamate, 80 mM MgSO4, 20 mM cysteine, 2 mM EGTA, and 80 mM NH_2_OH. The reaction was carried out at 37 °C for 30 min, before terminating by adding 0.5 mL of stop solution (370 mM FeCl_3_, 200 mM trichloroacetic acid, and 700 mM HCl). Following centrifugation at 5000 g for 15 min at 4 °C, the absorbance of the supernatant was measured at 540 nm. The unit of GS enzyme activity (U) expressed as μmol γ-glutamyl hydroxamate formed per gram fresh weight per minute (U = μmol·g^−1^·min^−1^ FW^−1^) based on a standard curve of γ-glutamyl hydroxamate. Each independent experiment consisted of three biological replicates and three independent experiments were performed to obtain the results.

#### 4.3.4. Measurement of Glutamate Synthase (GOGAT) Enzyme Activity

GOGAT activity was measured by monitoring the oxidation of NADH based on published methods [57]. Leaves harvested from plants aged 27, 31, and 39 days, respectively, and the leaf samples (0.5 g) were homogenized in an extraction buffer containing 100 mM Tris-HCl (pH 7.5), 1 mM EDTA, 1 mM MgCl_2_, and 10 mM 2-mercaptoethanol at 4 °C. The homogenate was centrifuged at 13,000 g for 15 min at 4 °C and the supernatant was collected for determining GOGAT activity. The reaction mixture consisted of 100 mM Tris-HCl (pH 7.5), 10 mM α-ketoglutarate, 10 mM L-glutamine, 0.2 mM NADH, and 200 μL enzyme extract in a total volume of 3 mL. The reaction was initiated by adding α-ketoglutarate, followed by monitoring the decrease in absorbance at 340 nm for 5 min at 30 °C. GOGAT activity was calculated using the extinction coefficient of NADH and expressed as μmol NADH oxidized per gram fresh weight per hour. Each independent experiment consisted of three biological replicates and three independent experiments were performed to obtain the results.

### 4.4. Confirmation of RNA-Seq Data by qRT-PCR

To confirm the reproducibility of the RNA-Seq data, 10 target genes were subjected to further qRT-PCR analysis. The Total RNA was extracted from 0.1 g ground leaf samples of 31-day-old seedlings using an RNA extraction kit (TransGen, Beijing, China) according to the manufacturer’s protocol. Then, RNA was transcribed into cDNA using a PrimScript RT kit (Takara, Kyoto, Japan). Quantitative real-time PCR (qRT-PCR) was performed with SYBR Premix Ex TaqTM (Takara, Kyoto, Japan) as the fluorescent dye, 100 ng cDNA as the template, and *Actin 2* as the internal reference. The primers for the target genes are listed in Appendix A. The relative expression of the target genes was calculated according to the 2^−ΔΔCT^ formula. Each sample consisted of three biological replicates, and three independent experiments were conducted to obtain the results.

### 4.5. RNA-Seq Analysis

The leaves of 31-day-old seedlings in the four groups (untreated control, BL-treated, Pyr-treated, and BL + Pyr treated) were used for RNA-Seq analysis, with three biological replicates per sample. Transcriptome sequencing was performed by Novogene Bioinformatics Technology Co., Ltd. (Tianjin, China) using an mRNA library preparation with unique molecular identifiers (UMIs) and the Illumina NovaSeq 6000 platform. Raw reads were filtered by removing adaptor sequences, reads containing poly-N sequences, and low-quality reads to obtain clean, high-quality data for subsequent analysis. Gene expression was quantified as fragments per kilobase of transcript per million mapped reads (FPKM). Differential expression analysis between samples was performed using the DESeq2 package (1.10.1), with a false discovery rate (FDR) adjusted *p*-values calculated by the Benjamini–Hochberg procedure. Genes with FDR < 0.05 were considered differentially expressed. Enrichment analysis of gene ontology terms and KEGG pathways among DEGs was conducted on the gene ontology (http://geneontology.org/) and KEGG PATHWAY (https://www.kegg.jp/kegg/pathway.html) databases, respectively. All raw sequence data from this study have been deposited into the NCBI’s SRA database with the link of https://www.ncbi.nlm.nih.gov/sra/PRJNA930055, under the accession number SAMN32982539 to SAMN32982550, the temporary Submission ID SUB12691837 and the BioProject ID PRJNA930055.

### 4.6. Metabolite Profiling

The leaves of 31-day-old seedlings in the four groups (untreated control, BL-treated, Pyr-treated, and BL + Pyr treated) were used for metabolite profiling, with three biological replicates per sample. Metabolites were extracted as previously described [58]. Quasi-targeted metabolite profiling was performed by LC-MS/MS using an ExionLC™ AD system (SCIEX) coupled with a TRAP^®^ 6500^+^ mass spectrometer (SCIEX) on Novogene Bioinformatics Technology Co., Ltd. (Tianjin, China). MRM (multiple reaction monitoring)-based screening detected metabolites matching the Novogene in-house database. The detected metabolites were quantified by peak integration and identified by Q3 and Q1 ions, RT (retention time), DP (depolymerization potential), and CE (collision energy). Peak integration and correction were performed in SCIEX OS v1.4. Differentially accumulated metabolites (DAMs) were determined by univariate *T*-test, with variable importance in projection (VIP) ≥ 1, fold change (FC) ≤ 0.8 or ≥1.2, and *p*-value < 0.05. These metabolites were annotated for KEGG pathways using the online websites: KEGG (http://www.kegg.jp/), HMDB (http://www.hmdb.ca/), and Lipidmaps (http://www.lipidmaps.org/) databases.

### 4.7. Integrated Transcriptome and Metabolome Analyses

The correlation between gene expression changes and metabolic alterations was determined by Pearson correlation analysis between the DEGs and DAMs identified by transcriptome and metabolomics analysis, respectively, using the Metware Cloud (https://cloud.metware.cn). Before conducting the correlation analysis, the data were log-transformed to improve the normality of the data distribution. A correlation coefficient < 0 indicates a negative correlation, while a correlation coefficient > 0 indicates a positive correlation. The common KEGG pathways shared by DEGs and DAMs information were obtained by mapping the genes and metabolites to the KEGG enrichment analysis in the NovoMajic Cloud Platform (https://magic.novogene.com/).

### 4.8. Statistical Analysis

To determine the statistical significance of the differences between treatments, *t*-test was performed using SPSS software (version 26.0, Chicago, IL, USA), and a *p*-value < 0.05 was considered to be statistically significant. Data are presented as the means ± standard deviation (SD) of three independent experiments.

## Figures and Tables

**Figure 1 ijms-24-16435-f001:**
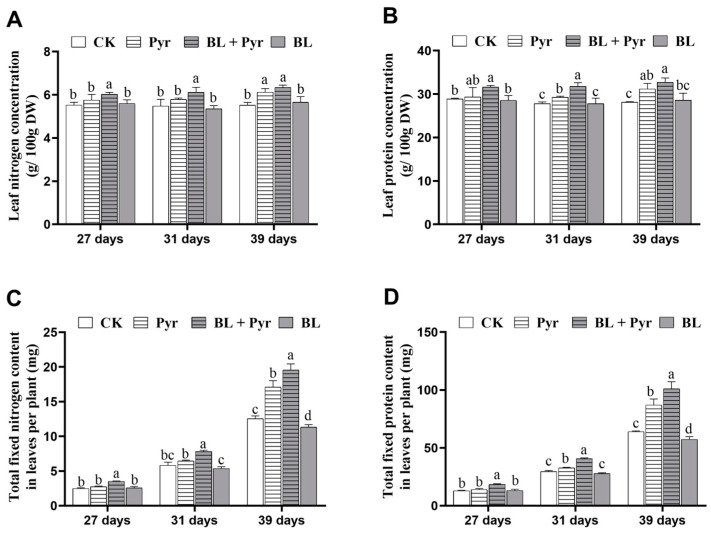
Determination of leaf nitrogen and protein content. The concentrations (%) of nitrogen (**A**) and protein (**B**) of leaves, and the total fixed weight (mg) of nitrogen (**C**) and protein (**D**) in leaves per plant. Leaves were sampled at 27-, 31-, and 39-day-old seedlings in the untreated group (CK), Pyr group (treated with 3 μM Pyr), BL + Pyr group (treated with 1 μM BL and 3 μM Pyr), and BL group (treated with 1 μM BL), respectively. Data are presented as the mean ± SD of three separate replicate experiments. Different letters indicate significant differences between the four treatments based on *t*-test comparisons at *p* < 0.05.

**Figure 2 ijms-24-16435-f002:**
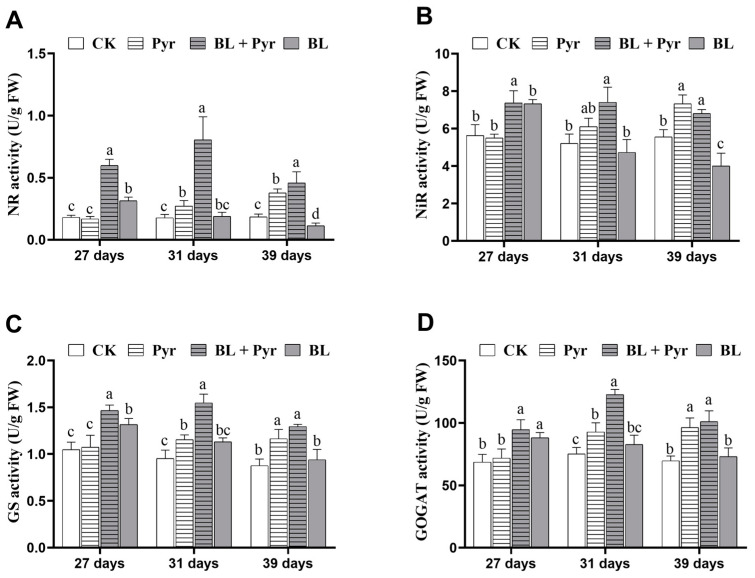
Activities of nitrogen assimilation key enzymes. The enzyme activities of NR (**A**), NiR (**B**), GS (**C**), and GOGAT (**D**). Leaves were sampled at 27-, 31-, and 39-day-old seedlings in the untreated group (CK), Pyr group (treated with 3 μM Pyr), BL + Pyr group (treated with 1 μM BL and 3 μM Pyr), and BL group (treated with 1 μM BL), respectively. Data are presented as the mean ± SD of three separate replicate experiments. Different letters indicate significant differences between the 4 treatments based on *t*-test comparisons at *p* < 0.05.

**Figure 3 ijms-24-16435-f003:**
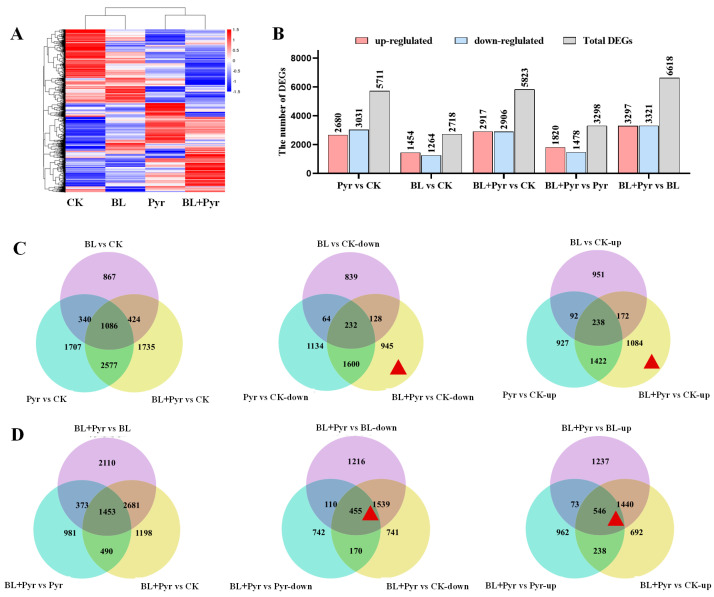
The identification of differentially expressed genes (DEGs) based on RNA-Seq: (**A**), hierarchical cluster analysis according to the FPKM of genes; (**B**), number of DEGs in separate comparison pairs; (**C**), Venn diagrams showing the overlapping and non-overlapping DEGs between the three groups (BL + Pyr, BL, and Pyr) and the untreated group; (**D**), Venn diagrams showing the overlapping and non-overlapping DEGs between the other three groups (untreated, BL, and Pyr groups) and the BL + Pyr group. The red triangle denoted the region that contained the DEGs specific to BL + Pyr.

**Figure 4 ijms-24-16435-f004:**
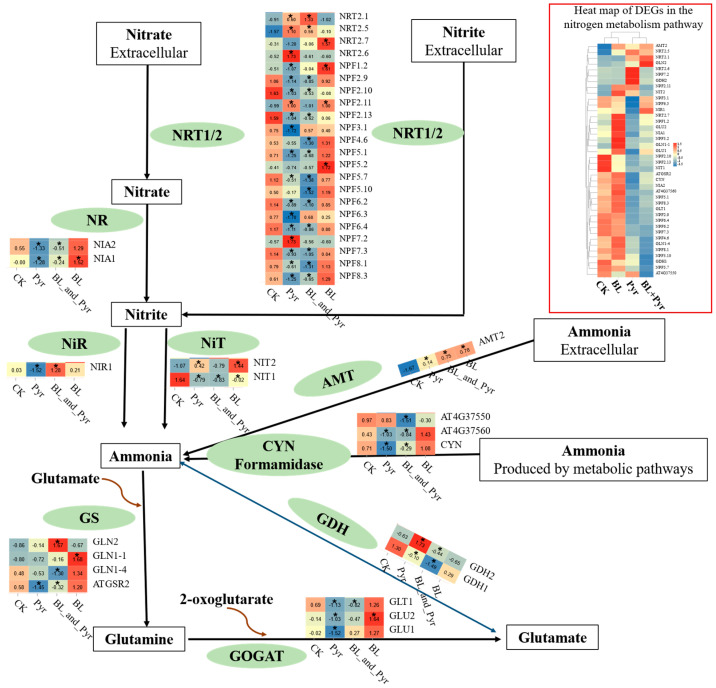
Nitrogen metabolism pathways tagged with DEGs. The heat map close to the enzyme (green elliptical box) showed the expression level of the gene encoding the corresponding enzyme. “*” in the heat map represented a significant difference of BL, Pyr, and BL + Pyr-treated group versus the untreated group, with redder colors in the heatmap representing higher expression.

**Figure 5 ijms-24-16435-f005:**
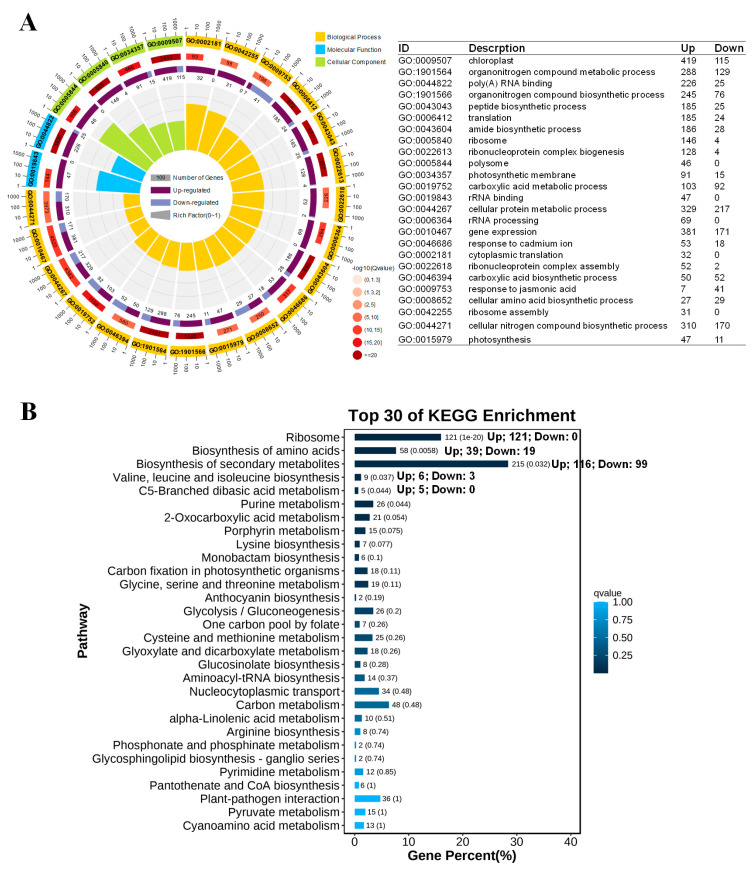
The GO and KEGG enrichment analysis of DEGs induced by BL + Pyr rather than by BL or Pyr. (**A**) Enrichment circular plot for the key significantly enriched GO terms of DEGs (the number of up and down DEGs in each GO term was represent at the right table); (**B**) bar graph of the top 30 significantly enriched KEGG pathways (The horizontal axis is the percentage of genes and the vertical axis is the details of KEGG terms, and the depths of the colors in the graph represent enrichment significance based on *p*-value, with *p*-value decreasing from dark to light. The number after the bars represented the number of DEGs, followed by the *p*-value in brackets).

**Figure 6 ijms-24-16435-f006:**
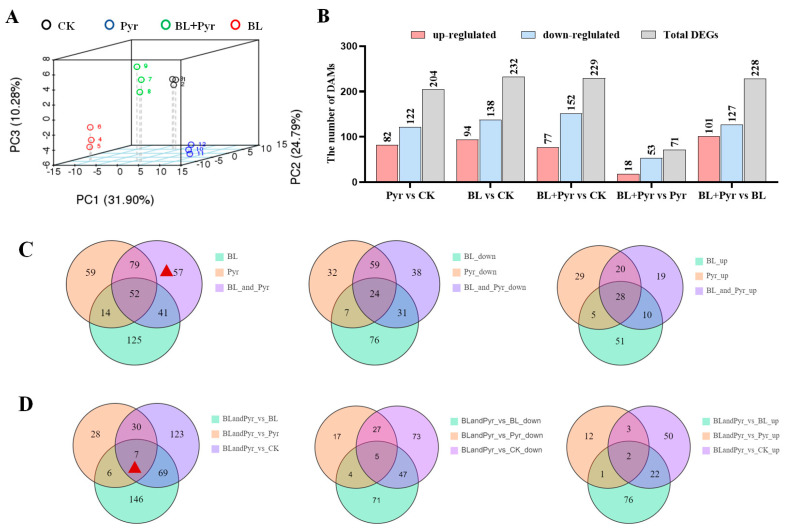
The identification of differentially accumulated metabolites (DAMs) based on metabolomic analysis. (**A**) PCA analysis according to the accumulation of metabolites in the leaves; (**B**) number of DAMs in separate comparison pairs; (**C**) Venn diagrams showing the overlapping and non-overlapping DAMs between the three groups (BL + Pyr, BL, and Pyr) and the untreated group; (**D**) Venn diagrams showing the overlapping and non-overlapping DAMs between the other three groups (untreated, BL and Pyr groups) and the BL + Pyr group. The red triangle denoted the region that contained the DAMs specific to BL + Pyr.

**Figure 7 ijms-24-16435-f007:**
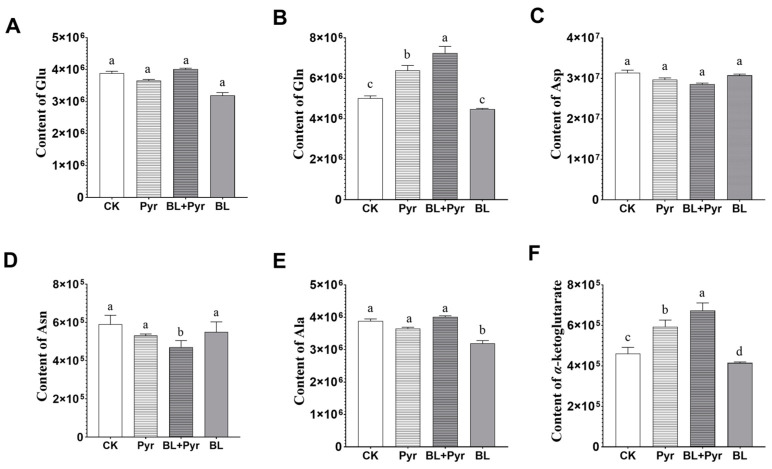
The accumulation of nitrogen metabolism pathway intermediates based on metabolomic analysis. The accumulation of Glu (**A**), Gln (**B**), Asp (**C**), Asn (**D**), Ala (**E**), and α-Ketoglutarate (**F**). Data are presented as the mean ± SD of three separate biological repetitions. Different letters indicate significant differences between 4 treatments based on *t*-test comparisons at *p* < 0.05.

**Figure 8 ijms-24-16435-f008:**
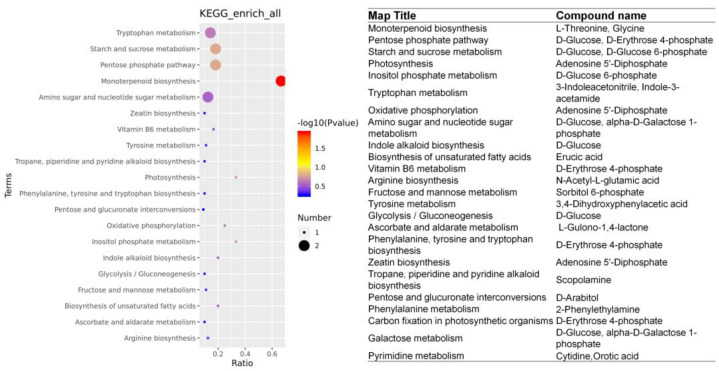
The KEGG enrichment analysis of metabolites differentially accumulated by BL + Pyr rather than by BL or Pyr. The left side was an enrichment bubble chart showing the enriched KEGG pathways, while the right side presented the BL + Pyr -unique DAMs annotated in the pathways.

**Figure 9 ijms-24-16435-f009:**
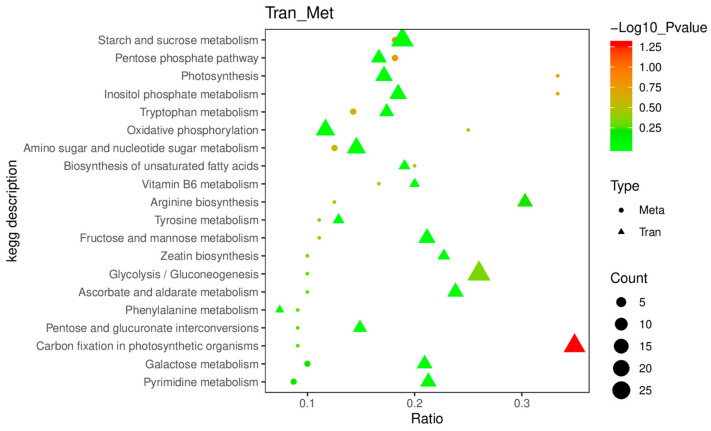
The co-enrichment analysis conducted on BL + Pyr-specific DEGs and DAMs within KEGG pathways. The enriched KEGG pathways are plotted along the X-axis, while the Y-axis signifies the enrichment factor. Triangles represent DEGs, circles represent DAMs. The bubble size correlates with the count of DAMs or DEGs, and the bubble color indicates the −log10 (*p*-value), providing a measure of statistical significance.

**Table 1 ijms-24-16435-t001:** The list of 133 BL + Pyr-specific transcription factors.

Family	ID	Name	Log_2_FC	Family	ID	Name	Log_2_FC
ARF	**AT1G19220**	**ARF19**	**−0.26**	MYB	AT1G09770	CDC5	0.12
	AT1G59750	ARF1	−0.17		**AT1G18570**	**MYB51**	**−0.41**
	**AT3G61830**	**ARF18**	**0.26**		AT1G18710	MYB47	0.78
ARR-B	AT2G01760	ARR14	−0.36		AT3G11450	-	0.22
	AT2G25180	ARR12	−0.26		AT4G21440	MYB102	0.60
	AT3G16857	ARR1	−0.19		**AT4G37260**	**MYB73**	**−0.71**
	AT4G31920	ARR10	−0.33		AT4G38620	MYB4	0.43
B3	AT1G01030	NGA3	0.89		AT5G06110	-	0.46
	AT1G16640	-	0.70	MYB-related	AT1G19000	-	−0.58
	AT2G36080	NGAL1	−0.63		AT1G70000	-	−0.57
	AT2G46870	NGA1	0.41		AT1G74840	-	−0.73
BBR/BPC	AT2G21240	BPC4	−0.19		AT3G21430	ALY3	−0.25
C2H2 zinc-finger protein	**AT1G24625**	**ZFP7**	**−0.54**		AT4G01280	RVE5	−0.44
	**AT1G27730**	**ZAT10**	**−0.52**		AT5G17300	RVE1	−1.09
	AT1G55110	IDD7	−0.36		AT5G45420	MAMYB	0.21
	AT1G72050	TFIIIA	0.24	NAC	**AT2G33480**	**NAC041**	**0.30**
	AT3G13810	IDD11	−0.41		**AT3G10500**	**NAC053**	**0.32**
	AT5G14140	-	−0.45		AT3G15500	NAC055	−1.74
	AT5G25160	ZFP3	−0.63		AT3G49530	NAC062	0.17
	AT5G52010	-	0.45		AT5G08790	NAC081	−0.43
C3H zinc-finger protien	AT3G02830	ZFN1	−0.18		AT5G24590	NAC091	0.23
	AT3G08505	MKRN	−0.41	RAV	**AT1G13260**	**RAV1**	**−1.17**
	AT3G55980	SZF1	−0.45		**AT1G25560**	**TEM1**	**−1.17**
	AT5G51980	-	0.27		**AT1G68840**	**RAV2**	**−0.79**
CO-like	AT3G02380	COL2	−0.48	SBP	AT1G20980	SPL14	−0.18
	AT5G57660	COL5	−0.11		AT1G53160	SPL4	−0.65
	AT5G59990	-	−0.58		**AT5G43270**	**SPL2**	**0.26**
DBB	AT1G06040	BBX24	−0.15	SRS	AT4G36260	SRS2	0.90
	AT4G38960	-	−0.41	TALE	AT4G36870	BLH2	−0.31
Dof	AT1G51700	DOF1.7	−0.30	TCP	AT1G72010	TCP22	−0.23
	AT3G47500	CDF3	−0.20		AT3G15030	TCP4	0.50
	AT5G60200	DOF5.3	−0.47		**AT3G47620**	**TCP14**	**−0.28**
ERF	AT1G33760	ERF022	−0.87		AT4G18390	TCP2	−0.26
	AT1G64380	ERF061	0.57	Trihelix	AT1G13450	GT−1	−0.25
	**AT2G20880**	**ERF053**	**0.77**		AT1G33240	AT-GTL1	−0.13
	**AT2G44940**	**ERF034**	**0.98**		AT5G28300	-	0.67
	**AT2G46310**	**CRF5**	**0.85**		AT5G63420	emb2746	0.64
	**AT3G15210**	**ERF4**	**−0.26**	WRKY	**AT1G69310**	**WRKY57**	**−0.40**
	AT4G06746	RAP2−9	−1.79		AT2G03340	WRKY3	−0.20
	**AT4G17490**	**ERF6**	**−0.41**		**AT2G38470**	**WRKY33**	**−0.41**
	**AT4G23750**	**CRF2**	**0.98**		**AT3G01970**	**WRKY45**	**−1.27**
	**AT5G44210**	**ERF9**	**0.64**		AT5G07100	WRKY26	−0.90
	**AT5G47230**	**ERF5**	**−0.40**		**AT5G13080**	**WRKY75**	**−0.70**
	AT5G53290	CRF3	0.68		AT5G45260	RRS1	−0.25
FAR13	AT3G06250	FRS7	0.28		AT5G52830	WRKY27	−0.54
G2-like	AT1G14600	-	−0.46	YABBY	AT1G08465	YAB2	−0.29
	AT1G25550	HHO3	−0.77	ZF-HD	AT2G18350	ZHD6	−0.39
	AT1G68670	HHO2	−0.83		**AT3G28920**	**ZHD9**	**0.37**
	AT3G46640	PCL1	−0.37	bHLH	AT1G06170	BHLH89	0.60
GATA	**AT3G24050**	**GATA1**	**0.24**		AT1G18400	BEE1	0.38
GRAS	AT1G07530	SCL14	−0.19		AT2G20180	PIF1	−0.30
	AT1G66350	RGL1	0.52		AT3G07340	BHLH62	−0.41
	**AT3G46600**	**SCL30**	**−0.66**		AT3G23690	BHLH77	−0.45
GeBP	AT4G00238	STKL1	0.43		AT4G29100	BHLH68	−0.38
	**AT4G25210**	**GEBPL**	**0.27**		AT4G30980	LRL2	0.63
HD-ZIP	**AT4G37790**	**HAT22**	**0.24**		AT5G38860	BIM3	−0.41
	**AT5G65310**	**ATHB5**	**0.16**		AT5G39860	PRE1	0.38
HSF	**AT3G51910**	**HSFA7A**	**−0.61**		AT5G50915	BHLH137	0.24
	AT5G03720	HSFA3	0.55		AT5G61270	BHLH72	−0.46
MIKC_MADS	AT2G45660	SOC1	−0.43	bZIP	**AT1G43700**	**VIP1**	**−0.23**
	**AT3G57230**	**AGL16**	**−0.25**		AT3G51960	BZIP24	−0.61
	AT5G15800	1-Sep	−1.09		AT3G62420	BZIP53	−0.41
PLATZ	AT4G17900	-	−0.20		AT4G02640	BZO2H1	−0.24
BTB/POZ	AT4G37610	BT5	−0.92		**AT4G36730**	**GBF1**	**0.20**
	AT5G63160	BT1	−1.19		**AT5G24800**	**BZIP9**	**−0.28**
	AT3G48360	BT2	−0.96		AT5G28770	BZIP63	−0.22
	AT5G67480	BT4	−0.83				

“log2FC” stands for “log2 fold change (the BL + Pyr group vs the untreated control)”. A positive log2FC indicated upregulation, while a negative value indicated downregulation. Bold font represented the transcription factor that has been reported to be implicated in N metabolism in the literature of Gaudinier et al., 2018 [49].

**Table 2 ijms-24-16435-t002:** The list of 57 BL + Pyr-specific DAMs.

Class	Compound Name	log_2_FC
Flavonoids and its derivatives	Idaein chloride	−0.50
	Tangeretin	−0.74
	Sakuranetin	−0.49
	Apigenin C-pentoside	−0.50
	O-methylnaringenin C-pentoside	−0.55
	(-)-Gallocatechin	−0.93
	Selgin C-hexoside	−0.45
	Methyl-Naringenin C-pentoside	−0.55
	Apigenin 4-O-rhamnoside	0.62
Carbohydrates and its derivatives	D-Glucose	−0.88
	alpha-D-Galactose 1-phosphate	0.28
	Sorbitol 6-phosphate	0.28
	D-Glucose 6-phosphate	0.33
	D-Erythrose 4-phosphate	0.55
	**L-Gulono−1,4-lactone**	**−1.43**
	D-arabitol	0.38
Amino acid and its derivatives	L-arginine	0.56
	N-Acetyl-L-glutamic acid	0.29
	Nepsilon-acetyl-L-lysine	−0.84
	Aminomalonic acid	0.38
	**Aspartic acid di-O-glucoside**	**1.92**
	D-asparagine	−0.33
Terpenoids	Dihydroartemisinic acid	−0.95
	**(-)-Menthol**	**−4.29**
	Artemether	−0.39
	(-)-trans-carveol	−0.68
	18-Nor−4,15-dihydroxyabieta−8,11,13-tried−7-one	0.84
Indoles and derivatives	**Indole-3-carboxaldehyde**	**−1.10**
	**3-indoleacetonitrile**	**−3.52**
	Indole-3-acetamide	0.49
Nucleotide and its derivates	Adenosine 5′-Diphosphate	0.81
	Cytidine	−0.98
	N-(9H-Purin−6-ylcarbamoyl)threonine	0.35
Alkaloids and derivatives	vasicine	−0.42
	Arecoline	−0.86
	**Dihydroxy tomatidine-O-hexosyl-O-rhamnoside**	**−1.16**
Amines	Methylguanidine	−0.53
	2-Phenylethylamine	−0.64
	Dl-Dihydrosphingosine	−0.38
Glycerolipids	Monopalmitin	−0.65
	MAG (18:4)	−0.84
Organic acid and its derivatives	**Scopolamine**	**−1.26**
	Adipic acid	0.55
Fatty acyls	Erucic acid	0.51
	Punicic acid	−0.58
Phenylpropanoids	Calceolarioside B	−0.63
	Isoscopoletin	−0.69
Benzoic acids and derivatives	3,4-Dihydroxyphenylacetic acid	−0.90
	**Methyl vanillate**	**−1.21**
Cinnamic acids and derivatives	**N-p-Coumaroyl putrescine**	**−1.16**
	N′-p-Coumaroyl agmatine	−0.52
Phenolic acids	2,6-Di-tert-butylphenol	−0.56
	**Denin**	**2.64**
Phospholipid	LysoPC 18:1	−0.73
Ethers	Ambroxane	0.58
Organoheterocyclic	Allantoin	0.29
Vitamins	**Orotic acid**	**−1.21**

“log_2_FC” stands for “log_2_ Fold Change (the BL + Pyr group vs the untreated control)”. A positive log2FC indicated an increase in accumulation, while a negative value indicated a decrease accumulation. Bold font represented the fold change greater than 2.

**Table 3 ijms-24-16435-t003:** The list of BL + Pyr-specific DAMs that were consistently differentially regulated by BL + Pyr versus the other three groups (untreated control, BL, and Pyr alone groups).

Name	Class	log_2_FC		
		BL + Pyr vs. CK	BL + Pyr vs. BL	BL + Pyr vs. Pyr
Monopalmitin	Glycerolipids	−0.65	−0.65	−0.67
Dihydroartemisinic acid	Terpenoids	−0.95	−0.95	−0.69
Indole−3-Carboxaldehyde	Indoles and derivatives	−0.99	−0.99	−0.58
D-Glucose	Carbohydrates And Its Derivatives	−0.88	−0.88	−0.63
(-)-Menthol	Terpenoids	−4.29	−4.29	−2.19
Indole−3-acetamide	Indoles and derivatives	0.60	0.60	0.82
L-arginine	Amino Acid And Derivatives	0.56	0.56	0.45

Notes: “log2FC” stands for “log2 Fold Change”. A positive log2FC indicated an increase accumulation, while a negative value indicated a decrease accumulation.

## Data Availability

Data are contained within the article.

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
