# Peer review of "Multi-Omics Analysis Reveals Synergistic Enhancement of Nitrogen Assimilation Efficiency via Coordinated Regulation of Nitrogen and Carbon Metabolism by Co-Application of Brassinolide and Pyraclostrobin in Arabidopsis thaliana"

_ijms, 2023, doi:10.3390/ijms242216435_

Round 1

Reviewer 1 Report

Comments and Suggestions for Authors

The work analyzed the effect of pyraclostrobin (active substance of fungicides) and brassinolide (plant hormone) used separatelly and together on nitrogen assimilation in Arabidopsis thaliana. The authors examined N content and protein content, the activities of key N assimilatory enzymes (nitrate reductase, nitrite reductase, glutamine synthetase, and glutamate synthase) as well as transcriptional changes, and metabolic changes. Research on the possibilities of improving nitrogen use efficiency is an important and current topic that can certainly gain readers' attention, taking into account agricultural sustainability and harmfull effects from excesive use of N fertilizers on environment. The manuscript is well written, some small suggestions for improving are below:

Lines 103 – 119 – in the end of the introduction section there should be clearly stated goal of the research rather than results

Line 652 – what kind of soil was used? Basic characteristic will be beneficial

Line 680 – Title of subsubsection 4.3.1 is missing

Sections 4.2. and 4.3. – give the age of plants (leaves) for specific analyses (similarly as it was done in line 751 for RNA-seq). Were the leaves taken from 27-, 31-or 39- days old plants or from all these days? This should be clearly stated in Material and methods too (not only in Results).

Lines 685, 715, „Add….” – convert these sentences so that it does not sound like recipe

Lines 614 and 772 – lack of space before references

Reviewer 2 Report

Comments and Suggestions for Authors

The manuscript deals with a relevant subject to International Journal of Molecular Sciences related with the nitrogen assimilation efficiency. The ms is very interesting, well written, with an interesting set of data and supported by deep and up-to-date literature and adequate discussion. However, there are some points, namely in materials and methods section, presented below, which require improvement and explanationI recommend that the ms should be accepted after minor revision.

Specific comments:

1.     Line 34: “Nitrate”. lower case.

2.      Line 42: NO2- is translocated. Correct.

3.     Results:  great part of the first paragraph should be removed. The information should be presented in material and methods section.

4.     Results: the terms protein and nitrogen fixation should be changed. Is fixation the correct designation?

5.     Figure 1: how authors explain some differences in letters between nitrogen concentration and protein concentration?

6.   Authors should provide more information about growing conditions, including if seedlings grown inside greenhouse or field conditions, in pots or soil, if in pots, the respective volume, the number of seedlings per pot or per soil area, the soil chemical and physical characteristics, the fertilization plan, the spray volume per seedling or per area and if one application or more applications…

Comments on the Quality of English Language

Minor editing of English language required.
